# Small-Molecule RAF265 as an Antiviral Therapy Acts against PEDV Infection

**DOI:** 10.3390/v14102261

**Published:** 2022-10-15

**Authors:** Jing Wang, Wen-Jun Tian, Cui-Cui Li, Xiu-Zhong Zhang, Kai Fan, Song-Li Li, Xiao-Jia Wang

**Affiliations:** 1Key Laboratory of Animal Epidemiology of the Ministry of Agriculture, College of Veterinary Medicine, China Agricultural University, Beijing 100193, China; 2Institute of Animal Sciences, Chinese Academy of Agricultural Sciences, Beijing 100193, China

**Keywords:** RAF265, PEDV, cytoskeleton, cellular translation, antiviral

## Abstract

Porcine epidemic diarrhea virus (PEDV), a member of the family *Coronaviridae*, causes acute diarrhea, vomiting, dehydration, and high mortality in newborn piglets, and has caused significant economic losses in the pig industry. There are currently no specific drugs available to treat PEDV. Viruses depend exclusively on the cellular machinery to ensure an efficient replication cycle. In the present study, we found that small-molecule RAF265, an anticancer drug that has been shown to be a potent inhibitor of RAF, reduced viral loads of PEDV by 4 orders of magnitude in Vero cells, and protected piglets from virus challenge. RAF265 reduced PEDV production by mediating cytoskeleton arrangement and targeting the host cell’s translation machinery. Treatment with RAF265 inhibited viral entry of PEDV S-glycoprotein pseudotyped viral vector particle (PEDV-pp), at half maximal effective concentrations (EC_50_) of 79.1 nM. RAF265 also presented potent inhibitory activity against viral infection by SARS-CoV-2-pp and SARS-CoV-pp. The present work may provide a starting point for further progress toward the development of antiviral strategies effective against coronavirus PEDV.

## 1. Introduction

Porcine epidemic diarrhea virus (PEDV) is an enveloped, single-stranded positive-sense RNA virus, which belongs to the genus *Alphacoronavirus* in the family *Coronaviridae* of the order *Nidovirales*. PEDV lacks its own translational apparatus and depends exclusively on its host’s translation machinery to efficiently synthesize viral proteins and product progeny virion. CoV mRNA generally undergoes cap-dependent translation with utilization of the eukaryotic translation initiation factor eIF4F complex [1]. Porcine epidemic diarrhea (PED) is caused by PEDV, with clinical symptoms including acute diarrhea, vomiting, dehydration, and a high mortality rate in newborn piglets [2]. The morbidity and mortality rates are almost 100% in suckling piglets [3]. Autogenous vaccines, commercialized vaccines, and intestinal content feeding do not efficiently control PED outbreaks, due to the hypervariability of PEDV, which makes field pandemics more heterogeneous [4,5]. There is an urgent need to develop effective antiviral therapies to compensate for the lack of vaccine immune protection.

Due to the intrinsically high mutation rate of RNA viruses, targeting host cellular machineries, which are essential for viral infection, is expected to be an advantageous strategy for antiviral therapeutics [6,7,8]. The current antiviral strategies for controlling viral infectivity focus on identification of agents capable of intervening in the essential steps for viral infection, including viral entry and replication.

Since many anticancer drugs have been established without toxicity in clinical trials, repurposing them for the use as antivirals may offer a more cost-effective and time-efficient approach to antiviral drug therapeutics [8]. RAF265 is a novel, orally active, small-molecule kinase inhibitor, and it is 14 times more selective for tumor cells than for non-tumor cells [9,10,11]. RAF265 is currently undergoing Phase 2 clinical trials for use in patients with locally advanced or metastatic melanoma without toxicity. In this study, we evaluated the efficacy of RAF265 as an antiviral agent.

We found that RAF265 significantly reduced virus proliferation in vitro and in vivo. Mechanistically, the antiviral effect of RAF265 is a dual inhibitory strategy targeting cellular translational machinery and cytoskeletal arrangement. Our findings suggest that repurposing RAF265 as an antiviral may offer a potential strategy to treat coronavirus infection.

## 2. Materials and Methods

### 2.1. Cells, Virus, Antibodies and Inhibitors

African green monkey kidney (Vero) cells were cultured in Dulbecco’s modified Eagle’s medium (DMEM, Gibco) supplemented with non-essential amino acids, 2 mM L-glutamine, sodium pyruvate, 10% heat-inactivated fetal bovine serum (FBS), 100 U/mL penicillin and 100 mg/mL streptomycin (all reagents were purchased from Gibco Invitrogen, Carlsbad, CA, USA) in a humidified 37 °C, 5% CO_2_ incubator. Porcine epidemic diarrhea virus (PEDV) strain CV777 was reproduced in Vero cells [12]. PEDV titers were determined by a 50% tissue culture infective dose (TCID_50_) assay in Vero cells [13]. Anti PEDV-N (1:1000) mouse monoclonal antibody was obtained from Alpha Diagnostic International (San Antonio, TX, USA). Antibodies to GAPDH (1:1000) and goat anti-mouse secondary antibodies conjugated to horseradish peroxidase (HRP, 1:10,000) were obtained from Beyotime Biotechnology (Shanghai, China). Rabbit polyclonal antibody to phosphorylated (p)-eIF4E (Ser 209, 1:1000) was purchased from Abcam (Cambridge, MA, USA). Rabbit polyclonal antibody to eIF4E (1:1000) was purchased from Proteintech (Wuhan, China). Antibodies to B-Raf, C-Raf, MNK1, p-Mnk1, p-S6K, p-4EBP1, EGFR, p-EGFR, p-FAK, Erk, p-Erk, Myosin, p-myosin, cofilin, p-cofilin and p-p38 (all antibodies diluted to 1:1000) were from Cell Signaling Technology (Danvers, MA, USA). The pharmacological inhibitors tested were from MedChemExpress (Monmouth Junction, NJ, USA).

### 2.2. Preparation of Cell Lysates and Western Blotting

The cells were collected by centrifugation, washed three times with pre-cooled PBS and dissolved in 200 μL lysis buffer in the presence of the protease inhibitor cocktail, and finally disrupted by sonication. The cell suspension was then fractionated by centrifugation at 10,000 rpm for 10 min at 4 °C. Solubilized proteins were harvested, electrophoresed in denaturing polyacrylamide gels, electroblotted onto a polyvinylidene fluoride (PVDF) membrane, and incubated with appropriate primary antibodies overnight at 4 °C. Protein bands were detected with secondary antibody conjugated to horseradish peroxidase (HRP) for 45 min at room temperature, and GAPDH was used as a loading control. 

### 2.3. Quantitative Real-Time PCR (qRT-PCR)

Total RNA was extracted with Trizol (Invitrogen). The cDNA was obtained using cDNA Synthesis SuperMix (TRANS). A two-step RT-PCR (SYBR Green I technology, Applied Roche) was performed using SYBR green super mix (Toyobo) according to the manufacturer’s protocol to measure transcription levels for several genes of interest. The primers used were as follows: PEDV-N: 5′- CTG GGT TGC TAA AGA AGG CG −3′ (forward), 5′- CTG GGG AGC TGT TGA GAG AA −3′ (reverse). GAPDH: 5′-GAT CAT CAG CAA TGC CTC CT −3′(forward), 5′- TGA GTC CTT CCA CGA TAC CA −3′(reverse). Relative fold changes were calculated following the 2^−ΔΔCT^ method. GAPDH was used as a control.

### 2.4. Confocal Microscopy Analysis

When Vero cells were infected with PEDV, RAF265 was added at the same time. After 1 h post-infection, the Vero cells were washed three times with pre-cooled PBS, and cells were fixed and incubated with primary antibody against PEDV-N overnight at 4 °C. Cells were washed and then incubated with TRITC-conjugated secondary antibodies. Nuclei were stained with DAPI. The cover slips were mounted on glass slides and cells were observed using a confocal laser scanning microscope. Experiments were conducted in duplicate in five independent wells.

### 2.5. Generation of Vero-eIF4E(S209A) Cells

Single guide RNA (sgRNA) was designed to target the area near a specific site using the online CRISPR design tool “http://crispr.mit.edu (accessed on 2 July 2018) ”, and a donor template was designed containing the mutation site. The sgRNA, donor template, and Cas9 plasmids were co-transfected into Vero cells and the cells were plated in 96-well plates by limit dilution to generate isogenic single clones. The clones were picked via screening by restriction endonuclease, and identified by Sanger sequencing. After screening, we obtained a total of 8 positive clones for extended culture. Mycoplasma test was performed with MycoAlert™ PLUS Mycoplasma Detection Kit of Lonza. Finally, the Ser was mutated to Ala at amino acid 209 of eIF4E. The sequence of the sgRNA was designed as follows: GCAGACACAGCTACTAAGAG (with 80.3% cleavage efficiency analyzed by on-line tool TIDE).

### 2.6. Production of Spike Pseudotyped Particles and Virus Entry Assay

To produce PEDV-pp, HEK293T cells were seeded 1 day prior to transfection in 10-cm plates and then transfected using Lipofectamine 2000. The plasmid DNA transfection mixture (1 mL) was composed of 15 µg of pNL-4.3-Luc-E−R− and 15 µg of pcDNA-PEDV-Spike. A non-enveloped lentivirus particle (Bald virus) was also generated as negative control. 16 h after transfection, the media was replaced with fresh media supplemented with 2% FBS. Supernatants containing PEDV-pp were typically harvested at 36–48 h after transfection and then filtered through a syringe filter (0.22 µm) to remove any cell debris. PEDV-pp was freshly used or allocated and frozen at −80 °C. SARS-CoV-2-pp and SARS-CoV-pp were constructed following the same method. 

To conduct the virus entry assay, Huh7 cells were seeded in 96-well plate at 1 day prior to transduction. The next day, 100 µL of supernatant containing PEDV-pp, SARS-CoV-2-pp, or SARS-CoV-pp was added into each well in the absence or presence of serially diluted RAF265. After 48 h of transduction, the cells were lysed in 100 µL of passive lysis buffer and 50 µL lysate was incubated with 100 µL of luciferase assay substrate according to the manufacturer’s instructions (Promega, Madison, WI, USA). Anti-coronavirus experiments usually use two to four duplicated wells for each treatment.

### 2.7. Statistics

All results were expressed as means and standard deviations (SD). Statistical analyses were performed using Prism 7.00 (GraphPad Software, La Jolla, CA, USA)). Significance was determined by one-way analysis of variance (ANOVA) with Dennett’s multiple-comparison test. Partial correlation analyses were evaluated using an unpaired Student’s *t*-test.

## 3. Results and Discussion

### 3.1. RAF265 Significantly Reduced Virus Proliferation In Vitro and In Vivo

To analyze the specificity of RAF265 for viral replication of PEDV, we investigated its impact on the levels of viral protein and viral yield in the monkey-kidney-derived cell line Vero. We observed a significant reduction in viral protein levels (PEDV-N) under treatment with RAF265 at 5 μM (Figure 1A). Furthermore, we found that viral yields were reduced by 4 orders of magnitude when treated with RAF265 at a concentration of 10 μM (Figure 1A). Of note, Vero cells are widely used for PEDV pathogen research and vaccine development. We showed that when not treated with RAF265 they demonstrated high uptake of PEDV (Figure 1A). We also evaluated the inhibitory effect of RAF265 on PEDV infection in 7-day piglets. As shown in Figure 1B, in piglets infected with PEDV, the epithelial cells of the mucosa were necrotic, the cytoplasm was vacuolated, the cytolysis disappeared, and the intestinal glands in the lamina propria were loosely arranged. After 2 doses of RAF265 at 25 mg/kg, the structure of mucosa, submucosa, muscularis and tunica adventitia were seen to be intact, and no obvious denatured necrosis in the tissue was observed in infected piglets. RAF265 also decreased viral mRNA level of PEDV in the blood and intestines (Figure 1C). It has been reported that PEDV-loaded RBCs can transfer the virus to CD3+ T cells, leading to intestinal infection via cell-to-cell contact [14], and also causes typical diarrhea symptom [15]. These results indicate that RAF265 reduces virus proliferation of PEDV in vitro and in vivo.

### 3.2. RAF265-Mediated Actin–Myosin Arrangement Interfered with PEDV Entry

We investigated which stages of PEDV infection were affected by RAF265. Laser confocal (Figure 2A) and qRT-PCR (Figure 2B) experiments showed that RAF265 interfered with PEDV entry into Vero cells. Especially, the absorption rate of virus particles at 4 °C for 1 h was measured in the absence or presence of different concentrations of RAF265. The result showed that RAF265 increased viral absorption (Figure 2B, top). Subsequently, we investigated viral endocytosis stage, and found that the expression levels of PEDV v-RNA were significantly reduced under treatment with RAF265 (middle, bottom). The significant reduction in the level of viral RNA in the cell indicates that RAF265 blocks the entry of PEDV into the Vero cell by targeting viral internalization.

Increasing evidence suggests that the choice of coronavirus entry mechanism is very complex. Coronavirus was initially thought to enter cells through direct fusion with the plasma membrane, but more recent evidence suggests that virus is able to enter cells through a receptor-mediated, clathrin- and caveola-independent endocytic pathway [16,17,18]. After PEDV reached the entry site, actin filaments of PEDV-infected cells retracted and concentrated around plasma membrane. Then, actin bundles aligned with plasma membrane for virus internalization [19]. At approximately 30–60 min post-infection, bound virus surfed toward the foot of filopodia via actin retrograde flow, while actin filament depolymerization occurred and transient blebs were formed on the cell surface [20]. Given that dynamic actin rearrangement is involved in viral absorption and endocytosis of coronavirus [21,22]. We sought to evaluate the effect of RAF265 on the actin cytoskeleton. 

In our study, the phosphorylation levels of cofilin (p-CFL), myosin light chain 2 (p-MLC2), and extracellular regulated protein kinase (p-Erk) were increased at 1 h under treatment with RAF265 in a dose-dependent manner (Figure 2C). Erk activation is required for PEDV replication [23]. Raf kinase inhibitors inhibit Erk phosphorylation in tumor cells but promote Erk phosphorylation in non-tumor cells [24]. It suggests that RAF265 presents antiviral activity during the early stage of PEDV infection by modulating p-cofilin; this event may be catalyzed by Erk activation. As reported, MLC2 plays an important role in the process by which virus surfs along filopodia on the host membrane [25] to the entry sites; the activation of MLC2 (p-MLC2) facilitates viral invasion [21]. Likewise, MLC2 may initiate the internalization of coronavirus by disintegrating F-actin to overcome the actin barrier [21,26]. Our findings suggest that RAF265 may mediate actin–myosin II network to interfere with the invasion of PEDV. The details need to be further investigated.

Different viruses exploit different aspects to co-opt Cdc42/Rac1-PAK signaling, to influence the behavior of infected cells, such as cytoskeletal rearrangements, antiapoptotic, and immune evasion [27]. We found that RAF265 inhibited the entry of PEDV into the cell, while Cdc42 inhibitor of ZCL278 [28] and ML141 [29] did not (Figure 2D). The present finding suggests that PEDV manipulates the cytoskeleton for virus entry in a Cdc42-independent manner, although PEDV [30] and another porcine coronavirus PHEV [20] rely on clathrin-mediated endocytosis for entry into cells, and PHEV utilizes Cdc42 to promote its own invasion [27,31]. Given these findings, the ability of RAF265 to regulate cytoskeletal arrangement should be further explored in antiviral drug development. 

### 3.3. RAF265 Blocked Viral Entry of Coronavirus PEDV, SARS-CoV, and SARS-CoV-2

To evaluate the ability of RAF265 treatment to inhibit viral entry by specifically blocking endocytosis, a recombinant PEDV S-glycoprotein pseudotyped viral vector particle (PEDV-pp), SARS-CoV-2-pp, and SARS-CoV-pp, an infectivity assay in Huh7 hepatocarcinoma cell line was performed. Note that pseudotyped particles are commonly used to study mechanisms of viral entry [32]. As shown in Figure 3, RAF265 presented potent inhibitory activity against viral infection by PEDV-pp, SARS-CoV-2-pp, and SARS-CoV-pp, at half maximal effective concentrations (EC_50_) of 79.1, 335.2, 469.9 nM, respectively. No measurable decrease in cell viability was detected below 10 μM in Huh7 cells (Figure 3). The potency of RAF265 highlights its potential utility as an effective entry inhibitor against SARS-CoV-2 and other zoonotic coronaviruses.

### 3.4. RAF265 Reduced the Synthesis of Viral Proteins of PEDV by Depressing p-eIF4E

Because RAF265 is a B-Raf kinase inhibitor, we explored whether the inhibitory effect of RAF265 on viral infections is related to the phosphorylation eIF4E (p-eIF4E), which is a downstream factor involved in the Ras/Raf/MAPK-regulated kinase signaling pathway. Vero-eIF4E (S209A) cell lines in which the phosphorylation-required residue serine 209 was substituted with alanine were established (eIF4E cannot be phosphorylated). The growth rate of the Vero-eIF4E (S209A) and the wild type Vero-WT cells was nearly consistent within 4 days after monolayer adherent culture. We observed that the S209A-Vero cells appear to be less susceptible than WT-Vero to viral infection (Figure 4A). Furthermore, upon viral infection by PEDV, the inhibitory effect of RAF265 on viral replication was more effective in the WT-Vero than in the S209A-Vero cells (Figure 4B). These results indicate that RAF265 effectively reduces the synthesis of viral proteins of PEDV by targeting p-eIF4E.

Viruses have been found to target the processes of eIF4E phosphorylation to facilitate selective translation for viral mRNAs during infection [33,34]. For example, the stimulation of eIF4E phosphorylation enhances viral translation and replication of Newcastle disease virus [34] and Herpes simplex virus 1 [35]. As the phosphorylation of eIF4E on serine 209 is required for proliferation of cancer but not normal cells [36]; inhibitors targeting p-eIF4E have therefore been considered for anticancer and antiviral therapeutics, in the hope that they would not produce excessive side effects. Here, we showed that small-molecule RAF265 depressed the level of p-eIF4E to exhibit antiviral activity against PEDV infections with relatively low likelihood of drug resistance.

Taken together, our findings suggest that RAF265 efficiently inhibits viral infection in vitro and in vivo, and mediates cytoskeleton arrangement and targets the host cell’s translation machinery (see schematic diagram in Figure 4C). Furthermore, RAF265-mediated cytoskeleton arrangement prevented the endocytosis of the coronaviruses PEDV, SARS-CoV and SARS-CoV-2. Altogether, our findings suggest that RAF265 may offer a starting point for the development of antivirals to treat coronavirus infection, and should be further investigated in clinical trials.

## Figures and Tables

**Figure 1 viruses-14-02261-f001:**
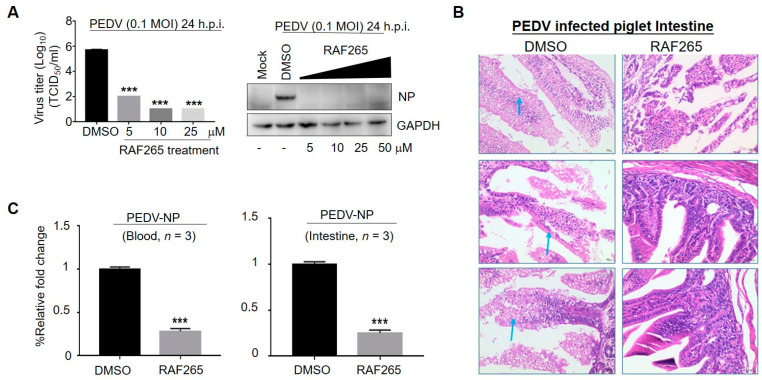
Inhibitory effect of RAF265 on PEDV infection in vitro and in vivo. (**A**) Vero cells were infected with PEDV at 0.1 MOI in the presence of the RAF265 at different concentrations for 24 h, viral yield was measured by TCID_50_ assay (left) and level of viral protein PEDV-N was analyzed by WB (right). (**B**,**C**) Six piglets were divided into two groups of three. One group was intraperitoneally infected with PEDV at 10^5^ TCID_50_, and another group was injected with PEDV in the presence of two doses of RAF265 (25 mg/kg) orally at an interval of 1 day. 7 days after infection, the piglets were killed and the Jejunum middle of intestine was harvested and fixed in 4% paraformaldehyde and embedded in paraffin, and then tissue sections were stained with hematoxylin and eosin (H&E) (**B**). Arrows indicate the epithelial cells of the mucosa were necrotic, the cytoplasm was vacuolated, the cytolysis disappeared, and the intestinal glands in the lamina propria were loosely arranged. Blood and intestine were harvested and total RNA was extracted using Trizol reagent, and then PEDV-NP mRNA was detected by qRT-PCR (**C**). h.p.i, hours post-infection; MOI, multiplicity of infection. Significance was evaluated using the t test (mean ± SEM, *n* = 3). *** *p* ≤ 0.001.

**Figure 2 viruses-14-02261-f002:**
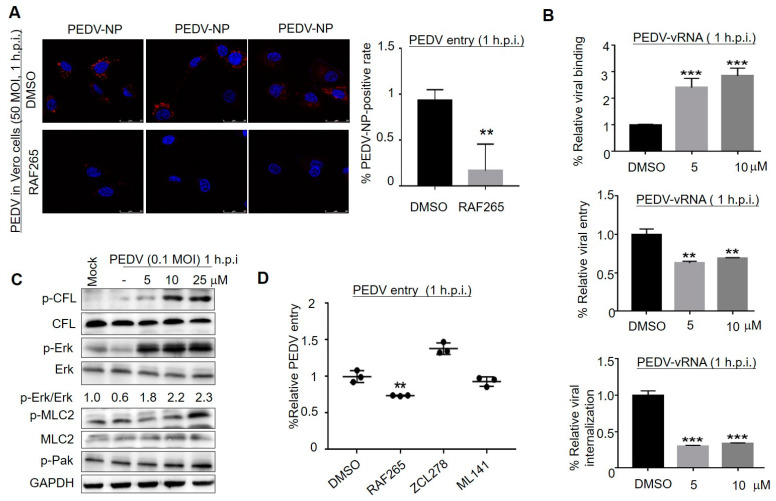
RAF265-mediated cytoskeleton arrangement involved in the early events of viral infection. (**A**) Infected cells were treated with RAF265 for 1 h at 37 °C, fixed, permeabilized, and stained for DAPI (blue) and PEDV NP (red). Viral entry was evaluated by confocal fluorescence microscopy magnification 200×. The experiments were performed three times and from 1 random horizon in each experiment. (**B**) Vero cells were infected with PEDV in the presence of DMSO or RAF265 for 1 h at 4 °C, washed three times with cold PBS, and then viral RNA was extracted for RT-PCR assay (top). Vero cells were infected with PEDV in the presence of DMSO or RAF265 for 1 h at 37 °C, washed three times with cold PBS. Viral RNA was extracted for RT-PCR assay (middle). Vero cells were infected with PEDV for 1 h at 4 °C, washed three times with cold PBS, and then treated with indicated concentrations of RAF265 for 1 h at 37 °C. Viral RNA was extracted for RT-PCR assay (bottom). (**C**) Vero cells were infected with PEDV at 0.1 MOI in the presence of RAF265 at different concentrations for 1 h. Cells were harvested for WB analysis with indicated antibodies. (**D**) Infected cells were co-treated with RAF265, ZCL278, or ML141 for 1 h at 37 °C and washed with cold PBS. Viral genome was extracted for qRT-PCR. h.p.i, hours post-infection; MOI, multiplicity of infection. Significance was evaluated using one-way ANOVA (mean ± SEM, *n* = 3). ** *p* ≤ 0.01; *** *p* ≤ 0.001.

**Figure 3 viruses-14-02261-f003:**
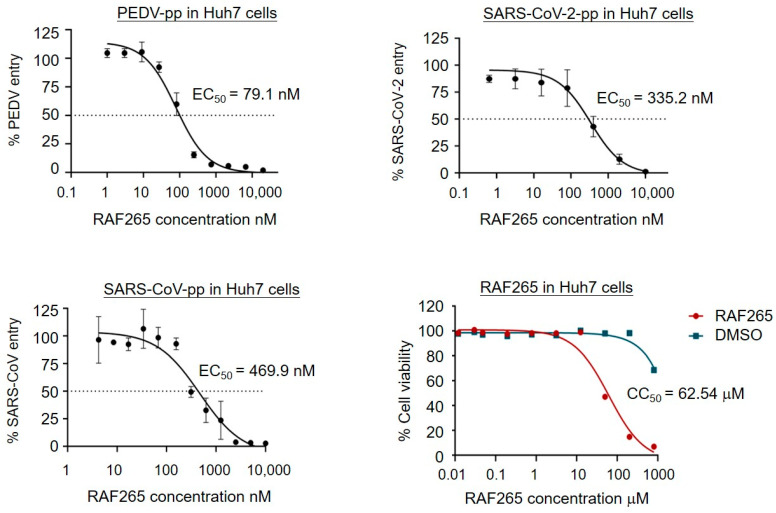
RAF265 targeted the entry of PEDV, SARS-CoV, and SARS-CoV-2. Huh7 cells were transduced with PEDV-pp, SARS-CoV-pp, or SARS-CoV-2-pp in the presence of serially diluted RAF265 or DMSO. Forty-eight hours after transduction, the cells were lysed and incubated with luciferase assay substrate. The cytotoxicity of RAF265 was evaluated by MTT assay. Data were analyzed with GraphPad Prism 7 (GraphPad Software, San Diego, CA, USA). For entry assay, the infectivity data were first inverted to antivirus activity. Each data set was normalized by the background control (no virus) to define the real value for 100%. The values shown in the graphs are presented as mean ± SEM.

**Figure 4 viruses-14-02261-f004:**
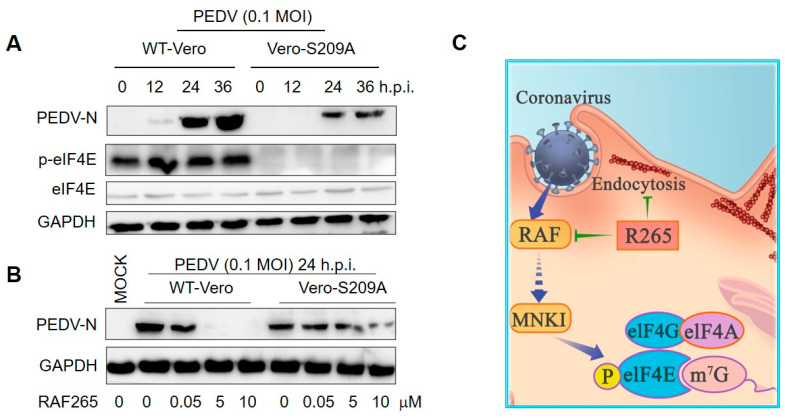
RAF265 suppressed the synthesis of viral proteins of PEDV. (**A**) Vero-WT and Vero-eIF4E (S209A) cells were challenged with PEDV for different times; cells lysates were harvested for WB analysis with indicated antibodies. (**B**) Vero-WT and Vero-eIF4E (S209A) cells were infected with PEDV in the presence of increasing doses of RAF265 for 24 h. Two cell lines were harvested for WB analysis with indicated antibodies. (**C**) Schematic representation of dual-targeting inhibitor RAF265 as active antiviral. After binding, PEDV enters into Vero cells by endocytosis, RAF265 mediates arrangement of actin cytoskeleton involved in the internalization of viral infection of PEDV. RAF265 also decreases the level of p-eIF4E to inhibit the synthesis of viral proteins. h.p.i, hours post-infection; MOI, multiplicity of infection.

## Data Availability

The data that support the findings of this study are available on request from the corresponding author.

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
