# Peer review of "Small-Molecule RAF265 as an Antiviral Therapy Acts against PEDV Infection"

_viruses, 2022, doi:10.3390/v14102261_

Round 1

Reviewer 1 Report

In this manuscript, the authors found one anti-cancer drug RAF265, inhibited PEDV infection in Vero cells and protected piglets from PEDV challenge. Viral vector particle assay showed that RAF265 inhibited viral entry of PEDV S-glycoprotein pseudotyped viral vector particle (PEDV-pp), SARS-CoV-2-pp, and SARS-CoV-pp. Generally, the content and structure of the manuscript are good. However, the following questions should be addressed before publication:

Major points:

1.      Line 105, please provide more detailed information about “screening by restriction endonuclease…..” It is not clear how clone screening was performed.

2.      FIG 2A, please provide the percentage of cells with PEDV-NP in DMSO and RAF265 treated cells.

3.      Line 184-185, the “and extracellular regulated protein kinase (p-Erk) increased at 1 h under treatment with RAF265 in a dose-dependent manner”…No increase was observed upon RAF265 treatment. Please provide the quantification or replace with new data.

4.      FIG 4A, please show the total eIF4E level.

5.      The trend of PEDV-N in FIG 1A is quite different from FIG 4B (samples of WT-Vero). Please provide quantities of RAF265 in FIG 4B.

Minor points:

1.      Line 184, “hour” instead of “h”.

2.      Please explain the abbreviation “h.p.i.” in the figure legend.

Reviewer 2 Report

In this study, the small molecule RAF265 as an antiviral therapy against PEDV infection, namely porcine epidemic diarrhea virus (PEDV), the authors found that the small molecule RAF265 has been shown to be a potent inhibitor of RAF, reducing the viral load of PEDV Reduces four orders of magnitude in Vero cells and protects piglets from viral attack. RAF265 reduces PEDV production by mediating cytoskeletal alignment and targeting the host cell's translation machinery. Treatment with RAF265 inhibits viral entry of PEDV S-glycoprotein pseudotyped viral vector particles (PEDV-pp) with a maximum effective concentration (EC50) of 79.1 nM. RAF265 also exhibited potent inhibitory activity against SARS-CoV-2-pp and SARS-CoV-pp viral infections. It was then concluded that this work may provide a starting point for further development of effective antiviral strategies against coronavirus PEDV.

Comments:

1.      Fig. 1C., Blood and intestine were harvested and total RNA was extracted using Trizol reagent, and then PEDV-NP mRNA was detected by qRT-PCR. PEDV is mainly contagious in the lung and intestinal system, please provide a support that it is also infectious in the blood system.

2.      Figure 1B., What is HE, full name and explanation required. What do the arrows indicate? This should be stated in the legend.

3.      Figure 2A. Cells without fluorescence are required for comparison. Why are there so few infected cells with red fluorescence, suggesting that host cells are fairly recalcitrant to viral infection even without RAF265 treatment?

4.      Figure 2A., the authors should point out the differences between these 6 subfigures. Because they are all represented by PEDV-NP at the top.

5.      Figure 2B. The authors noted that confocal and qRT-PCR experiments showed that RAF265 interfered with PEDV entry into Vero cells (Figure 2A, 2B). However, Figure 2A,B are confocal results, not qRT-PCR experiments.

6.      Figure 2B. Why is the viral adsorption rate increased in RAF265-treated cells?

7.      Figure 2D. (D) Infected cells were co-treated with RAF265, ZCL278 or ML141 for 1 hr at 37°C and washed with cold PBS. The viral genome was extracted for qRT-PCR analysis. In these experiments, the authors indicate that the viral genome is extracted from the incoming PEDV virus, but how do the authors exclude viruses that adhere to the cell surface?

8.      Figure 4. In Figure 2C. the authors found that the level of p-CFL was affected by RAF265 treatment, however, this gene is not shown in the cartoon in Figure 4?
